# Research on Comprehensive Evaluation Model of a Truck Dispatching System in Open-Pit Mine

Xiangyu Kou [1,2], Xuebin Xie [1] , Yi Zou [3], Qian Kang [1,4,*] and Qi Liu [1,2]

1   School of Resources and Safety Engineering, Central South University, Changsha 410083, China; kxy@cimr.com.cn (X.K.); xbxie@csu.edu.cn (X.X.); liuqicsy@163.com (Q.L.)
2   Changsha Mining Research Institute Co., Ltd., Changsha 410012, China
3   Zijin Mining Group Co., Ltd., Xiamen 572000, China; zou.yi@zijinmining.rs
4   School of Emergency Management and Safety Engineering, Jiangxi University of Science and Technology, Ganzhou 341000, China
*   Correspondence: kangqianray@126.com

**Abstract:** In this paper, a comprehensive evaluation factor model of a truck dispatching system in open-pit mines is constructed from the three dimensions of optimal route, traffic flow planning, and real-time dispatching, and the final combined weight of the factor is determined according to game theory. On this basis, a comprehensive evaluation model of a truck dispatching system in open-pit mines based on gray relational analysis—technology for order preference by similarity to an ideal solution (GRA-TOPSIS) is established. Taking the truck dispatching system in five open-pit mines as the research background, the advantages and disadvantages of the dispatching system were comprehensively evaluated, and the differences between the dispatching systems were analyzed using the radar chart method. The research shows that the evaluation results of the comprehensive evaluation model of the truck dispatching system in open-pit mines based on GRA-TOPSIS are in line with the reality, which is more conducive to analyzing and comparing the advantages and disadvantages of the systems, effectively identifying the differences of various systems, and making the evaluation of truck dispatching systems more scientific. The research results of this paper broaden the evaluation of truck dispatching systems and provide a theoretical basis for the optimization of truck dispatching systems.

**Keywords:** truck scheduling; game theory; gray correlation; approximate ideal solution; blind number theory



## 1. Introduction

"Eco", "safe", "intelligent", and "efficient" are the requirements for the sustainable development of the mining industry. China is rich in mineral resources, and open-pit mining is the most commonly used mining method. In recent years, due to the continuous expansion of the scale of open-pit mining and greater investment in large-scale equipment, open-pit mining has gradually developed in the direction of mechanization, automation, and intelligence, which has greatly improved the mining efficiency of mineral resources and improved the economic benefits of mining enterprises. Truck transportation has outstanding advantages, such as strong environmental adaptability, as well as flexible and convenient scheduling. Therefore, it is still the most common transportation method used in open-pit mines in China, and the transportation cost of open-pit mining accounts for more than 50% of the production cost of the entire mine [1]. For this reason, strengthening the management of transportation equipment and improving the working efficiency of transportation equipment has become the key for mining enterprises to control costs. With the continuous development and application of new-generation information technologies such as big data, artificial intelligence technology, and the Internet of things, as well

as the continuous advancement of intelligent mine construction, more and more large-scale open-pit mines have established truck dispatching systems. The establishment of a truck dispatching system fully realizes the optimal planning, optimal design, optimal management, and optimal control of the mine, and also conforms to the requirements of the new era. In the production and transportation process of open-pit mines, the dispatching system can dispatch vehicles in real time according to real-time data, which can significantly improve transportation efficiency; moreover, it can save and control operating costs for enterprises [2]. In the truck dispatching system, it is necessary to make the truck freight the lowest and the waiting time the shortest at the same time. In the transportation equipment route planning, the optimal route aiming at the shortest transportation distance and the lowest transportation consumption can better meet the actual needs of the mine. Although the open-pit mine truck dispatching system can fully improve the benefits of mining enterprises in general, it is particularly urgent for mining enterprises to establish a suitable, efficient, and economic dispatching system and continuously improve the intelligence level of the dispatching system. Accordingly, for mining enterprises, it is of great practical significance and value to carry out the evaluation of open-pit mine truck dispatching systems and continuously improve the functions of the dispatching system, as well as continuously promote the optimization and upgrading of the system.

Many experts and scholars have studied the optimal design and comprehensive evaluation of the truck dispatching system in open-pit mines, achieving a series of research results, which have greatly promoted the intelligent development of truck dispatching systems. In terms of truck dispatching system optimization, some algorithms improved the economy of the operation of the dispatching system, such as the artificial fish swarm neural network optimization algorithm [3], improved quantum particle swarm optimization algorithm based on genetic algorithm [4], and combined optimization algorithm from simulated annealing algorithm and genetic algorithm [5]. In terms of truck dispatching system evaluation, Zhao Bin et al. constructed a matter–element extension comprehensive evaluation model for truck dispatching systems in open-pit mines and calculated weight factors using the weighting method [6]. Furthermore, using the combined weighting method, which realized the dynamic evaluation of the truck dispatching system in open-pit mines, on the basis of establishing an optimization factor system for truck dispatching in open pit mines, Zhao Songsong et al. used the improved AHP to analyze the weights of optimization factors [7]. The research shows that road network conditions, operation time, and transportation factors have the greatest impact on the truck scheduling process, which provides theoretical guidance for the research on truck scheduling in open-pit mines. Through comparison, it was found that several studies have been conducted on the optimization of the truck dispatching system, while only few studies have been conducted on the comprehensive evaluation of the truck dispatching system. Therefore, a relatively mature theoretical system, a perfect evaluation factor system, and an effective comprehensive evaluation model have not yet been formed, making it difficult to truly guide practice. The optimization of the truck dispatching system in open pit mines mainly focuses on three aspects: route optimization, traffic flow planning, and real-time dispatching. Besides, the influencing factors involved in each optimization aspect are different. Therefore, after in-depth research on the theory and process of truck scheduling in open pit mines, a comprehensive evaluation model for truck dispatching systems in open-pit mines is built in this paper. The three aspects of optimal route, traffic flow planning, and real-time dispatching have been taken into consideration in this model. The evaluation of the truck dispatching system in open-pit mines is a multi-attribute decision problem, with the need to consider many factors that are incompatible. Furthermore, qualitative factors are mainly used, which increases the difficulty of evaluation to a certain extent. In the research of multi-attribute decision-making problems, the technique for order of preference by similarity to ideal solution (TOPSIS) is a relatively widely used theoretical model. It evaluates the pros and cons by calculating the closeness of each decision plan to the positive and negative ideal solutions, with the advantages of relatively simple calculation and strong

operability. The TOPSIS method has been widely used in system feasibility evaluation [8], system risk assessment [9], alternative optimization [10], and so on, achieving satisfactory effects. However, studies have shown that the traditional TOPSIS method only considers the Euclidean distance between factors when making decisions, and it is difficult to truly and objectively reflect the overall dynamic trend [11]. Gray relational analysis (GRA) is a model for systematic decision making based on the gray relational degree [12]. Therefore, a GRA-TOPSIS comprehensive evaluation model was established by coupling these two theoretical models: TOPSIS and GRA. The GRA-TOPSIS comprehensive evaluation model can give full play to the advantages of the GRA model and make up for the shortcomings of TOPSIS, thus making the decision more scientific and reliable. When applying the GRA-TOPSIS model for comprehensive evaluation and decision making, a systematic and comprehensive multilevel comprehensive evaluation factor system needs to be established first, before calculating the factor weights. The rationality and objectivity of the factor weights directly affect the reliability of the evaluation results. Weight calculation methods are mainly divided into two categories: subjective weighting methods and objective weighting methods. Subjective weighting methods mainly rely on the subjective experience of the evaluator, ignoring the true reflection of the objective facts; hence, the weight has strong subjectivity. On the other hand, objective weighting methods are based on the attributes of the factor data, which can fully reflect their essential characteristics. However, objective weighting methods ignore the subjective initiative of people, and they are not ideal for application to situations with poor or limited information. Therefore, purely adopting subjective or objective weighting methods is not conducive to comprehensive evaluation results [13,14].

In this paper, an evaluation model based on GRA-TOPSIS was established in order to improve the comprehensive evaluation of the truck dispatching system. Through this evaluation model, the existing truck dispatching system can be evaluated. In addition, the existing system can be optimized, upgraded, and improved according to the evaluation results, which can further improve the production efficiency and management level of open-pit mining enterprises, and further control the operating costs of enterprises. By constructing a comprehensive evaluation factor system of the truck dispatching system, the G1 method, the improved CRITIC method, and game theory were used in this paper to determine the final weights of factors. Since the evaluation factors were qualitative, the assignment of factors was mainly based on expert scoring. Considering the subjective blindness and uncertainty of experts in scoring, it was difficult to directly give an exact scoring standard. Therefore, blind number theory was introduced. The blind number matrix was constructed to process the factor scores, further reducing the influence of subjective factors on the evaluation results. Lastly, taking the truck dispatching system of five open-pit mines as the research background, a comprehensive evaluation of the truck dispatching system was carried out to verify the applicability of the model.

## 2. Construction of Comprehensive Evaluation Factor System for Truck Dispatching System in Open-Pit Mines

The open-pit mine truck dispatching system is based on wireless transmission technology, relies on GPS satellites, and takes the mine database as the core. It realizes traffic flow planning and real-time dynamic scheduling of trucks by seeking the optimal running route of trucks under the constraints of transportation volume and stripping ratio. Therefore, the truck dispatching system evaluation is closely related mainly to optimal routes, traffic flow planning, and real-time dispatching. Sixteen factors, including transportation speed and road network nodes, were selected to construct a comprehensive evaluation factor system for the truck dispatching system in open-pit mines. These factors were chosen in adherence to three principles: an objective and comprehensive evaluation system, measurable data, and representative and typical factors. Furthermore, on the basis of previous research results [6,7], various opinions from evaluation experts of truck dispatching systems in open-pit mines and system development, as well as from optimization personnel and on-site

technicians, were fully taken into consideration when selecting factors. The evaluation results were divided into four grades: excellent, good, moderate, and poor. Table 1 presents the selection of factors and the criteria for the classification of factors.

**Table 1.** Structure and classification of factor system.

| First-Level Factor | Second-Level Factor | Factor Grading Standard | | | |
| | | Excellent | Good | Moderate | Poor |
| --- | --- | --- | --- | --- | --- |
| | Transportation speed $x_{11}$ | (90–100) | (80–90) | (60–80) | (0–60) |
| | Road network node $x_{12}$ | (90–100) | (80–90) | (60–80) | (0–60) |
| Optimal route $X_1$ | Road quality $x_{13}$ | (90–100) | (80–90) | (60–80) | (0–60) |
| | Road slope $x_{14}$ | (90–100) | (80–90) | (60–80) | (0–60) |
| | Road distance $x_{15}$ | (90–100) | (80–90) | (60–80) | (0–60) |
| | Road capacity constraints $x_{21}$ | (90–100) | (80–90) | (60–80) | (0–60) |
| | Shovel capacity constraints $x_{22}$ | (90–100) | (80–90) | (60–80) | (0–60) |
| Traffic flow planning $X_2$ | Unloading capacity constraints $x_{23}$ | (90–100) | (80–90) | (60–80) | (0–60) |
| | Ore grade constraints $x_{24}$ | (90–100) | (80–90) | (60–80) | (0–60) |
| | Production Plan $x_{25}$ | (90–100) | (80–90) | (60–80) | (0–60) |
| | Truck capacity $x_{26}$ | (90–100) | (80–90) | (60–80) | (0–60) |
| | Traffic continuity $x_{27}$ | (90–100) | (80–90) | (60–80) | (0–60) |
| | Key projects $x_{31}$ | (90–100) | (80–90) | (60–80) | (0–60) |
| Real-time scheduling $X_3$ | System priority $x_{32}$ | (90–100) | (80–90) | (60–80) | (0–60) |
| | Scheduling capability $x_{33}$ | (90–100) | (80–90) | (60–80) | (0–60) |
| | Scheduling efficiency $x_{34}$ | (90–100) | (80–90) | (60–80) | (0–60) |

*Note: The first column "Comprehensive evaluation factor system of truck dispatching system "O" in open-pit mine" spans all rows.*

## 3. Theoretical Basis

### 3.1. Blind Number Theory

In the 1990s, academic Wang Guangyuan first proposed blind number theory, which became an ideal tool for analyzing and dealing with uncertain information, gray information, and blind information. Its specific definition is presented below [15,16].

Assuming that $G$ is an interval-type gray number set, $\alpha_i \in G$, $f(x)$ is a gray function on $G$. If $\alpha_i \in (0, 1)$ $(i = 1, 2, \ldots, n)$, and

$$f(x) = \begin{cases} \alpha_i, and\ x = \alpha_i (i = 1, 2, \ldots n) \\ 0,\ otherwise \end{cases}. \tag{1}$$

When $i \neq j$, $x_i \neq x_j$, and $\Sigma_{i=1}^n \alpha_i = \alpha \leq 1$, the function $f(x)$ is called a blind number. It is represented by $\{[x_1, x_n], f(x)\}$, where $x_1$ is the lower limit of $x$, and $x_n$ is the upper limit of $x$. Hence, $\alpha_i$ is the credibility of $x$ in $x_i$, and $\alpha = \Sigma_{i=1}^n \alpha_i$ is the total credibility of $x$.

### 3.2. Factor Weight

3.2.1. G1 Method to Calculate the Subjective Weight of Factors

The analytic hierarchy process (AHP) is currently the most frequently used method of subjective weighting. In this method, experts combine their subjective experience to provide the relative importance ratios of all factors, and then construct a comparison matrix to calculate the weights of factors. In the calculation process, this method relies too much on the subjective experience of experts; especially when analyzing and processing problems with a huge evaluation factor system and complex factor relations, it is often necessary to repeatedly adjust the comparison matrix to meet the normalization requirements. As a result, the weight calculation process is relatively complicated, and the weight is subjective. On this basis, Guo Yajun proposed the G1 method, which is an improved method of the traditional AHP. In the process of weight calculation in the G1 method, it is only necessary to compare the relative importance of two factors and determine the relative importance ratio. Factor weights are determined on the basis of the relative importance. Compared with the AHP method, the G1 method not only reduces the requirements for the

evaluator's subjective cognition, but also does not need to go through the consistency check process, which makes the calculation process simpler and achieves satisfactory application results [17]. The specific calculation process is described below.

(1) Determine the factor sequence. In order to evaluate the object, it is assumed that $n$ evaluation factors are selected. After all the evaluators are fully discussed and a consensus is formed, the most important factor is selected from the $n$ factors, denoted as $X_1$, and the weight is denoted as $w_1$. Then, the most important factor is selected from the remaining $n - 1$ factors until all $n$ factors are selected according to their relative importance in sequence, yielding the factor sequence $(X_1, X_2, \ldots, X_n)$.

(2) Determine the relative importance ratio of factors. The relative importance ratio $r_j$ is determined by the evaluator according to the factor sequence $(X_1, X_2, \ldots, X_n)$. Table 2 presents a description of the relative importance factor. The relative importance ratio $r_j$ is calculated using Equation (2).

$$r_j = \frac{w_j}{w_{j-1}} j = 2, 3, \cdots n. \tag{2}$$

(3) The $n$-th factor weight is calculated using Equation (3).

$$w_n = \left(1 + \sum_{k=2}^{n} \prod_{j=k}^{n} r_j\right)^{-1}. \tag{3}$$

(4) The remaining $n - 1$ factor weights are calculated using Equation (4).

$$w_{j-1} = r_j w_j. \tag{4}$$

**Table 2.** Values of relative importance ratio.

| $r_j$ | Relative Importance | $r_j$ | Relative Importance |
|---|---|---|---|
| 1.0 | Equally important | 1.2 | Slightly more important |
| 1.4 | Obviously more important | 1.6 | Significantly more important |
| 1.8 | Extremely more important | 1.1, 1.3, 1.5, 1.7, 1.9 | Situations between the above description |

### 3.2.2. Improved CRITIC Method to Determine Objective Weight of Factors

The criteria importance through intercriteria correlation (CRITIC) method is an objective weighting method proposed by Diakoulaki based on the attributes of the data. This method assigns the weights of factors on the basis of two criteria: data contrast strength and data conflict. Data contrast strength is determined using the standard deviation, while data conflict is characterized by the correlation coefficient. This preserves the correlations between data to the greatest extent. In practical application, due to the different dimensions and magnitudes of factors, the data cannot be directly compared and analyzed; therefore, the original CRITIC method needs to be improved. In this paper, the coefficient of variance was introduced to eliminate the differences in the dimension and magnitude of the factors, further improving the reliability of the weighting results. The specific calculation process is described below [18].

(1) Build the original evaluation matrix. Assuming that $n$ evaluation factors are selected and $m$ objects are evaluated, the original evaluation factor matrix $X$ is constructed as follows:

$$X = \begin{bmatrix} x_1(k_1) & x_2(k_1) & \cdots & x_n(k_1) \\ x_1(k_1) & x_2(k_2) & \cdots & x_n(k_2) \\ \vdots & \vdots & \vdots & \vdots \\ x_1(k_m) & x_2(k_m) & \cdots & x_n(k_m) \end{bmatrix} = (x_j(k_i))_{n \times m}, \ j = 1, 2, \ldots, n, \ i = 1, 2, \ldots, m. \quad (5)$$

(2) Normalize original evaluation matrix. The matrix $X$ is normalized on the basis of the Z-score, and the normalized matrix $X^*$ is obtained. The normalization formula is as follows:

$$x_j^*(k_i) = \frac{x_j(k_i) - \overline{x_j}}{s_j}, \quad (6)$$

where $\overline{x_j}$ is the mean of the $j$-th factor, and $s_j$ is the standard deviation of the $j$-th factor.

(3) Calculate the coefficient of variation. In order to compare the factors more conveniently, the coefficient of variation is introduced. The coefficient of variation is calculated using Equation (7).

$$v_j = \frac{s_j}{\overline{x_j}}, \quad (7)$$

where $v_j$ is the coefficient of variation of the $j$-th factor.

(4) Determine the coefficient of independence. The correlation coefficient matrix of the normalized matrix $X^*$ is determined using the Pearson's coefficient of correlation, and the independence coefficient between the factors is determined using the correlation coefficient matrix.

$$\sum_{q=1}^{n} (1 - \rho_{q1}), \sum_{q=1}^{n} (1 - \rho_{q2}), \ldots, \sum_{q=1}^{n} (1 - \rho_{qm}). \quad (8)$$

(5) Calculate objective weights. The comprehensive coefficient is a coefficient that directly reflects the amount of information contained in the factor, thus determining the weight of the factor. The formula for calculating the comprehensive coefficient $h_j$ is as follows:

$$h_j = v_j \sum_{q=1}^{n} (1 - \rho_{qm}), \quad (9)$$

where $h_j$ is the comprehensive coefficient of the $j$-th factor.

The factor weights are calculated using Equation (10) on the basis of the improved CRITIC method.

$$W_j = \frac{h_j}{\sum_{q=1}^{n} h_j}, \quad (10)$$

where $W_j$ is the weight of the $j$-th factor.

### 3.2.3. Calculation of Final Factor Weights Using Game Theory

According to game theory, selecting $N$ kinds of weight calculation methods to form a set of weights $W = \{w_1, w_2, \cdots, w_N\}$, for any linear combination of these $N$ vectors, $w = \sum_{k=1}^{N} a_k w_k^T$ is obtained. According to the optimal strategy, dispersion minimization is performed on $w$ and $w_k$.

$$Min \| \sum_{k=1}^{N} a_k w_k^T - w_j^T \|^2, j = (1, 2, \cdots, N). \quad (11)$$

According to the properties of differentiation of the matrix, it can be known that the optimal first-order derivative condition satisfies the following formula:

$$\begin{bmatrix} w_1 w_1^T & \cdots & w_1 w_N^T \\ \vdots & & \vdots \\ w_N w_1^T & \cdots & w_N w_N^T \end{bmatrix} \begin{pmatrix} a_1 \\ \vdots \\ a_N \end{pmatrix} = \begin{bmatrix} w_1 w_1^T \\ \vdots \\ w_N w_N^T \end{bmatrix}. \tag{12}$$

The coefficients $(a_1, a_2, \dots , a_N)$ are normalized to obtain the optimal weight coefficient, and the final weight is determined as follows:

$$w^* = \sum_{k=1}^{N} a_k^* w_k^T. \tag{13}$$

*3.3. Improved GRA-TOPSIS Evaluation Method*

3.3.1. Traditional TOPSIS Method

The principle of TOPSIS is to calculate the Euclidean distance between the factor of the object to be evaluated and the positive ideal solutions, as well as the Euclidean distance between the factor of the object to be evaluated and the negative ideal solutions, before determining the degree of closeness to realize the ranking of the pros and cons of the alternatives. A greater degree of closeness indicates a better alternative. The specific calculation process is described below [19].

(1) Build a multi-attribute decision matrix. Assuming that there are $m$ alternatives for selection and $n$ evaluation factors are selected, the multi-attribute decision matrix A can be constructed as

$$A = \begin{bmatrix} x_{11} & x_{12} & \cdots & x_{1n} \\ x_{21} & x_{22} & \cdots & x_{2n} \\ \vdots & \vdots & \vdots & \vdots \\ x_{m1} & x_{m2} & \cdots & x_{mn} \end{bmatrix} \tag{14}$$

(2) Normalize the decision matrix. Considering the difference in dimensions between the factors, the factors need to be normalized, whereby the benefit factor is normalized using Equation (15), and the cost factor is normalized using Equation (16).

$$x_{ij} = \frac{x_{ij} - minx_{ij}}{maxx_{ij} - minx_{ij}} \tag{15}$$

$$x_{ij} = \frac{maxx_{ij} - x_{ij}}{maxx_{ij} - minx_{ij}}. \tag{16}$$

(3) Build a weighted standardized decision matrix. The weighted standardized decision matrix is developed according to the standardized matrix and the factor weights. The calculation formula is as follows:

$$C = \begin{bmatrix} w11x11 & w12x12 & \cdots & w1nx1n \\ w21x21 & w22x22 & \cdots & w2nx2n \\ \vdots & \vdots & \vdots & \vdots \\ wm1xm1 & wm2xm2 & \cdots & wmnxmn \end{bmatrix} \tag{17}$$

where $w$ is the factor weight, determined using game theory in this paper.

(4) Calculate the closeness of the object. The ideal solution is determined as follows:

$$\begin{cases} C^+ = \left\{ \left( \max_i x_{ij} | x \in X_1 \right), \left( \min_i c_{ij} | x \in X_2 \right) \right\} \\ C^- = \left\{ \left( \min_i x_{ij} | x \in X_1 \right), \left( \max_i c_{ij} | x \in X_2 \right) \right\} \end{cases}, \tag{18}$$

where $C^+$ represents the positive ideal solution set, and $C^-$ represents the negative ideal solution set; $X_1$ represents the benefit-type factor set, and $X_2$ represents the cost-type factor set.

The Euclidean distance between each alternative and the ideal solution is calculated as follows:

$$\begin{cases} D_i^+ = \sqrt{\sum_{j=1}^{m} (x_{ij} - C_j^+)^2} \\ D_i^- = \sqrt{\sum_{j=1}^{m} (x_{ij} - C_j^-)^2} \end{cases}, \tag{19}$$

where $d_i^+$ represents the Euclidean distance between each alternative and the positive ideal solution, and $d_i^-$ represent the Euclidean distance between each alternative and the negative ideal solution.

The closeness of each alternative to the positive ideal solution is calculated as follows:

$$B_i = \frac{D_i^-}{D_i^- + D_i^+}. \tag{20}$$

Finally, the alternatives are ranked according to the calculation results of the closeness of each alternative. The pros and cons of the objects to be evaluated can also be obtained.

### 3.3.2. Improved TOPSIS Method

The traditional TOPSIS method uses the Euclidean distance to represent the closeness of each object to the positive and negative ideal solutions to judge the pros and cons of the object. Studies have shown that, when multiple objects to be evaluated are located on the mid-perpendicular line of the ideal solution, the closeness obtained on the basis of the Euclidean distance algorithm is equal, so they cannot be compared. In addition, this method also ignores the correlation between factors. Therefore, this method often cannot truly and objectively reflect the pros and cons of the object to be evaluated. GRA is one of the core contents of grey system theory. It can describe and evaluate the state of the system over a period of time semi-qualitatively and semi-quantitatively by analyzing the known information, and then evaluate the overall level of the system. it is especially suitable for an evaluation environment with less data and fuzzy gray information. Accordingly, this paper uses the GRA method to improve the traditional TOPSIS method. By calculating the gray correlation coefficient, a new formula for calculating closeness is created to achieve a more reasonable and effective distinction between the pros and cons of the objects to be evaluated. The specific process is described below [11].

(1) Calculate weighted normalization matrix. The calculation formula of the weighted normalization matrix is shown in Equation (17). According to the calculation results, the optimal set of factors $U_j^*$ can be determined, which is then used as the reference sequence for improving GRA-TOPSIS.

$$U_j^* = (R_0^*(1), R_0^*(2), \ldots, R_0^*(m)), \tag{21}$$

where $R_0^*(j)$ is the optimal factor value in each object to be evaluated.

(2) Calculate the gray correlation coefficient $s_{ij}$ of each factor. After determining the gray coefficient of each factor, the gray correlation coefficient matrix $S = (s_{ij})_{m \times n}$ can be obtained. The calculation formula of the gray correlation coefficient is

$$s_{ij} = \frac{\min_i \min_j \Delta_i(j) + \zeta \max_i \max_j \Delta_i(j)}{\Delta_i(j) + \zeta \max_i \max_j \Delta_i(j)}, \tag{22}$$

where $\Delta_i(j) = \left| R_0^*(j) - r_{ij} \right|$, and $\zeta$ is the resolution coefficient, with a range of 0–1 (generally 0.5).

(3) Determine the positive ideal solution $s_0{}^+$ and negative ideal solution $s_0{}^-$ of the matrix *S*. The calculation formula is as follows:

$$s_0^+ = \max_{\substack{i \\ 1 \leq i \leq n}} s_i(j) = \left( s_0^+(1), s_0^+(2), \ldots, s_0^+(m) \right),$$ (23)

$$s_0^- = \min_{\substack{i \\ 1 \leq i \leq n}} s_i(j) = \left( s_0^-(1), s_0^-(2), \ldots, s_0^-(m) \right),$$ (24)

where $s_i(j)$ is the gray correlation coefficient of the *j*-th factor of the *i*-th object to be evaluated.

(4) Calculate the Euclidean distance using Equation (25).

$$\begin{cases} d_i^+ = \sqrt{\sum_{j=1}^{m} \left( s_i(j) - s_0^+(j) \right)^2} \\ d_i^- = \sqrt{\sum_{j=1}^{m} \left( s_i(j) - s_0^-(j) \right)^2} \end{cases}.$$ (25)

(5) Calculate the relative closeness of the gray association using Equation (26).

$$G_i = \frac{d_i^-}{d_i^+ + d_i^-}.$$ (26)

The calculation results of the relative closeness of the gray relation are ranked in order of size to obtain the pros and cons of the object to be evaluated.

*3.4. The Realization Process of Comprehensive Evaluation Model Based on Game Theory Using GRA-TOPSIS*

As mentioned above, this paper established a comprehensive evaluation model of GRA-TOPSIS based on a combined weighting method using game theory. The specific calculation process was as follows: (1) by analyzing the influencing factors of the object to be evaluated, a comprehensive evaluation factor system for the truck dispatching system in open-pit mines was constructed, and factors were classified into different levels; (2) experts gave the factor scoring interval of each object to be evaluated, and then according to blind number theory, the factor assignment of each object to be evaluated was calculated, and the G1 method, the improved CRITIC method, and game theory were applied to calculate the subjective weight, objective weight, and combined weight of factors, respectively; (3) the factor correlation function was constructed, and the factor correlation at each level was calculated; (4) the comprehensive closeness was calculated, according to which a comprehensive evaluation of the object to be evaluated was conducted, and the pros and cons of the object to be evaluated were determined.

## 4. Model Application

In order to verify the applicability of the GRA-TOPSIS model based on combined weighting using game theory in the comprehensive evaluation of the truck dispatching systems in open-pit mines, five mines *(K₁–K₅)* were taken as the research objects, and the calculation was carried out according to the principle of the comprehensive evaluation model. Five experts were organized to score the systems according to the system construction, actual operation, and relevant system design data, combined with the scoring criteria.

*4.1. Data Processing*

Since each evaluation factor was a qualitative factor, the assignment of each factor was determined by expert scoring. Considering the subjective randomness and uncertainty in

the direct scoring of experts, in order to better reflect and evaluate the actual situation of the dynamic changes in the truck dispatching system in open-pit mines, each expert was required to give a scoring interval for each factor. In addition, considering the differences in the cognitive level of experts, the concept of "expert credibility" was introduced as a function of their academic rank, professional title, academic degree, working years, and other factors. The credibility and comprehensive credibility of each expert are shown in Table 3.

**Table 3.** Basic information of experts.

| No | Academic Rank and Professional Title | Working Years | Academic Degree | Credibility | Comprehensive Credibility |
|---|---|---|---|---|---|
| $S_1$ | Associate professor | 6 | Doctor | 0.85 | 0.195 |
| $S_2$ | Professor | 7 | Doctor | 0.90 | 0.207 |
| $S_3$ | Senior engineer | 5 | Master | 0.85 | 0.195 |
| $S_4$ | Engineer | 8 | Master | 0.85 | 0.195 |
| $S_5$ | Senior engineer | 12 | Master | 0.90 | 0.207 |

Mine $K_1$ was taken as an example to illustrate the calculation principle of blind number theory. The factor scores of mine $K_1$ given by the experts are shown in Table 4.

**Table 4.** Scoring results of experts in mine $K_1$.

| Evaluation Factors | Expert Scoring Results | | | | |
|---|---|---|---|---|---|
| | $S_1$ | $S_2$ | $S_3$ | $S_4$ | $S_5$ |
| $X_{11}$ | (85–96) | (88–95) | (92–95) | (90–98) | (88–94) |
| $X_{12}$ | (80–85) | (82–88) | (78–88) | (75–86) | (82–89) |
| $X_{13}$ | (70–78) | (72–77) | (75–80) | (77–82) | (75–85) |
| $X_{14}$ | (70–75) | (70–78) | (72–75) | (70–80) | (68–73) |
| $X_{15}$ | (81–88) | (85–90) | (88–95) | (75–88) | (84–90) |
| $X_{21}$ | (75–78) | (75–80) | (80–82) | (77–83) | (75–88) |
| $X_{22}$ | (80–85) | (85–92) | (85–95) | (78–83) | (85–98) |
| $X_{23}$ | (81–84) | (78–85) | (75–89) | (73–86) | (85–92) |
| $X_{24}$ | (82–94) | (85–93) | (90–94) | (88–92) | (90–96) |
| $X_{25}$ | (85–95) | (88–95) | (92–98) | (88–95) | (91–97) |
| $X_{26}$ | (80–85) | (78–83) | (77–84) | (74–88) | (75–89) |
| $X_{27}$ | (81–85) | (85–93) | (85–96) | (78–86) | (85–95) |
| $X_{31}$ | (78–84) | (80–88) | (78–89) | (75–87) | (86–93) |
| $X_{32}$ | (80–84) | (78–88) | (77–86) | (75–88) | (75–85) |
| $X_{33}$ | (75–85) | (78–84) | (76–85) | (75–83) | (74–83) |
| $X_{34}$ | (83–87) | (85–94) | (85–95) | (78–88) | (88–93) |

Since the experts' scoring intervals overlapped with the factor grading standard of the open-pit mine truck dispatching system, it was necessary to reconstruct the scoring interval as a grading standard. On the basis of the factor grading interval in Table 1, combined with blind number theory, the expert scoring intervals were used to construct a blind number matrix. Taking factor $X_{11}$ as an example, the experts determined the scoring intervals to be (85–96), (88–95), (92–95), (90–98), and (88–94). After rearrangement, the scoring intervals without crossover were obtained as (85–88), (88–90), (90–92), (92–94), (94–95), (95–96), and (96–98). Combined with the comprehensive credibility of experts, the credibility of the rearranged scoring interval was calculated as $\theta_1 = (88 - 85)/(96 - 85) \times 0.195 = 0.0532$. Similarly, the credibility of other scoring intervals was $\theta_2 = 0.1638$, $\theta_3 = 0.2126$, $\theta_4 = 0.3426$, $\theta_5 = 0.1384$, $\theta_6 = 0.0437$, and $\theta_7 = 0.0457$. According to the reliability of the recalculated score interval, the blind number function was built as follows:

$$f_{X_{11}}(x) = \begin{cases} 0.0532 & 85 \leq x < 88 \\ 0.1638 & 88 \leq x < 90 \\ 0.2126 & 90 \leq x < 92 \\ 0.3426 & 92 \leq x < 94 \\ 0.1384 & 94 \leq x < 95 \\ 0.0437 & 95 \leq x < 96 \\ 0.0457 & 96 \leq x < 98 \end{cases}.$$

According to the obtained blind number function, the intervals were divided according to the truck dispatching system level. The possibility of the evaluation factor $X_{11}$ falling in four intervals was calculated. The final blind number matrix function was obtained as follows:

$$f'_{X_{11}}(x) = \begin{cases} 0 & 0 \leq x \leq 60 \\ 0 & 60 \leq x \leq 80 \\ 0.2170 & 80 \leq x \leq 90 \\ 0.7830 & 90 \leq x \leq 100 \end{cases}.$$

Using the same method to process factors $X_{12}$–$X_{34}$, the final blind number matrix $D$ was obtained as follows:

$$D = \begin{bmatrix} 0 & 0 & 0.2170 & 0.7830 \\ 0 & 0.1277 & 0.8723 & 0 \\ 0 & 0.8185 & 0.1815 & 0 \\ 0 & 1.000 & 0 & 0 \\ 0 & 0.0901 & 0.7706 & 0.1393 \\ 0 & 0.5798 & 0.4202 & 0 \\ 0 & 0.0780 & 0.8157 & 0.1063 \\ 0 & 0.1756 & 0.7356 & 0.0888 \\ 0 & 0 & 0.3312 & 0.6688 \\ 0 & 0 & 0.2894 & 0.7106 \\ 0 & 0.2825 & 0.7175 & 0 \\ 0 & 0.0732 & 0.7166 & 0.2102 \\ 0 & 0.1819 & 0.6998 & 0.1183 \\ 0 & 0.2528 & 0.7472 & 0 \\ 0 & 0.2675 & 0.7325 & 0 \\ 0 & 0.0390 & 0.6941 & 0.2669 \end{bmatrix}.$$

According to Table 1 and the principle of the interpolation method, the insertion points of each graded interval were determined as $a_1 = 30$, $a_2 = 70$, $a_3 = 85$, and $a_4 = 95$, and the final evaluation score of each factor was obtained. Taking factor $X_{11}$ as an example, according to the calculation of the blind number matrix, it can be known that the possibilities of factor $X_{11}$ falling into the four grading intervals were 0, 0, 0.2170, and 0.7830, respectively. Therefore, the final score of factor $X_{11}$ could be calculated as $x_{11} = 0 \times 30 + 0 \times 70 + 0.2170 \times 85 + 0.675 \times 95 = 92.83$. The final scores of the other factors and other objects to be evaluated are shown in Table 5.

**Table 5.** Factor assignment of objects to be evaluated.

| Factor | Mine | | | | |
|--------|------|------|------|------|------|
| | $K_1$ | $K_2$ | $K_3$ | $K_4$ | $K_5$ |
| $x_{11}$ | 92.83 | 87.68 | 78.79 | 84.56 | 75.37 |
| $x_{12}$ | 83.08 | 81.51 | 84.77 | 75.46 | 77.79 |
| $x_{13}$ | 72.72 | 75.86 | 74.96 | 79.52 | 81.38 |
| $x_{14}$ | 70.11 | 75.11 | 74.29 | 76.45 | 85.46 |
| $x_{15}$ | 85.04 | 83.96 | 81.57 | 80.45 | 83.28 |

**Table 5.** *Cont.*

| Factor | Mine | | | | |
|---|---|---|---|---|---|
| | $K_1$ | $K_2$ | $K_3$ | $K_4$ | $K_5$ |
| $x_{21}$ | 76.3 | 74.12 | 78.28 | 81.36 | 85.74 |
| $x_{22}$ | 84.89 | 83.15 | 81.29 | 80.75 | 84.39 |
| $x_{23}$ | 83.25 | 81.59 | 83.19 | 78.56 | 79.37 |
| $x_{24}$ | 91.69 | 85.37 | 81.39 | 78.54 | 88.25 |
| $x_{25}$ | 92.11 | 87.46 | 75.49 | 74.26 | 86.37 |
| $x_{26}$ | 80.76 | 78.56 | 74.39 | 75.78 | 81.34 |
| $x_{27}$ | 86.18 | 81.58 | 79.48 | 82.57 | 84.88 |
| $x_{31}$ | 83.45 | 78.52 | 76.31 | 75.73 | 78.79 |
| $x_{32}$ | 81.21 | 80.15 | 84.13 | 78.15 | 82.39 |
| $x_{33}$ | 80.99 | 81.45 | 78.59 | 74.63 | 82.34 |
| $x_{34}$ | 87.08 | 88.14 | 81.75 | 78.59 | 84.51 |

*4.2. Factor Weight Calculation*

As mentioned above, this paper used the G1 method, improved CRITIC method, and game theory to calculate the subjective weight, objective weight, and combined weight of the factors, respectively. The specific calculation results are described below.

(1)　Calculate the subjective weights of factors using the G1 method.

The comprehensive evaluation factor system of the truck dispatching system in the open-pit mine was constructed, and then experts determined each factor sequence set and the relative importance ratio. Taking the first-level factor layer as an example, experts gave the factor sequence $X_2 > X_3 > X_1$, with the relative importance ratios of $r_1 = 1.8$ and $r_2 = 1.6$. According to Equations (2)–(4), we can get $w_3 = (1 + 1.8 \times 1.6 + 1.6) - 1 = 0.183$, $w_2 = 0.183 \times 1.6 = 0.291$, and $w_1 = 0.291 \times 1.8 = 0.526$. Thus, the weights of the first-level factors were (0.183, 0.526, 0.291). In the same way, the weights of secondary factors could be obtained as $W_1 = 0.072, 0.034, 0.021, 0.018, 0.038, 0.100, 0.043, 0.039, 0.132, 0.120, 0.029,$ $0.063, 0.050, 0.042, 0.104,$ and $0.095$.

(2)　Calculate the objective weights of factors using the improved CRITIC method.

According to Equations (5)–(10) and Table 5, a comprehensive decision matrix was obtained. After normalizing the data of the comprehensive decision matrix, the difference coefficients and the independence coefficients were calculated to obtain the final objective weights of the factors. The difference coefficient calculation results were 0.074, 0.043, 0.041, 0.066, 0.020, 0.051, 0.020, 0.024, 0.055, 0.085, 0.035, 0.029, 0.035, 0.025, 0.035, and 0.042, while the objective weights of the factor were $W_2 = 0.137, 0.074, 0.098, 0.134, 0.019, 0.112, 0.018,$ $0.040, 0.051, 0.083, 0.036, 0.036, 0.037, 0.043, 0.036,$ and $0.046$.

(3)　Calculate the final weights of factors using game theory.

According to the weight calculation results, there was a big difference between the subjective weights $W_1$ and the objective weights $W_2$, indicating that the single weighting method was either subjective or objective, with a significant impact on the final evaluation result. Therefore, the weights were combined and optimized according to game theory to achieve balanced weight calculation results. According to Equations (11)–(13), MATLAB software was applied to calculate the combined weight coefficients, obtaining $a = 0.5442$ and $b = 0.5988$. After normalizing the combined weight system, the final combined weight coefficients were $a^* = 0.476$ and $b^* = 0.524$. Thus, the final factor combination weights were $W = 0.106, 0.055, 0.061, 0.079, 0.028, 0.106, 0.030, 0.039, 0.090, 0.100, 0.0325, 0.049, 0.043,$ $0.042, 0.068,$ and $0.069$.

The relative importance of each factor is shown in Figure 1.

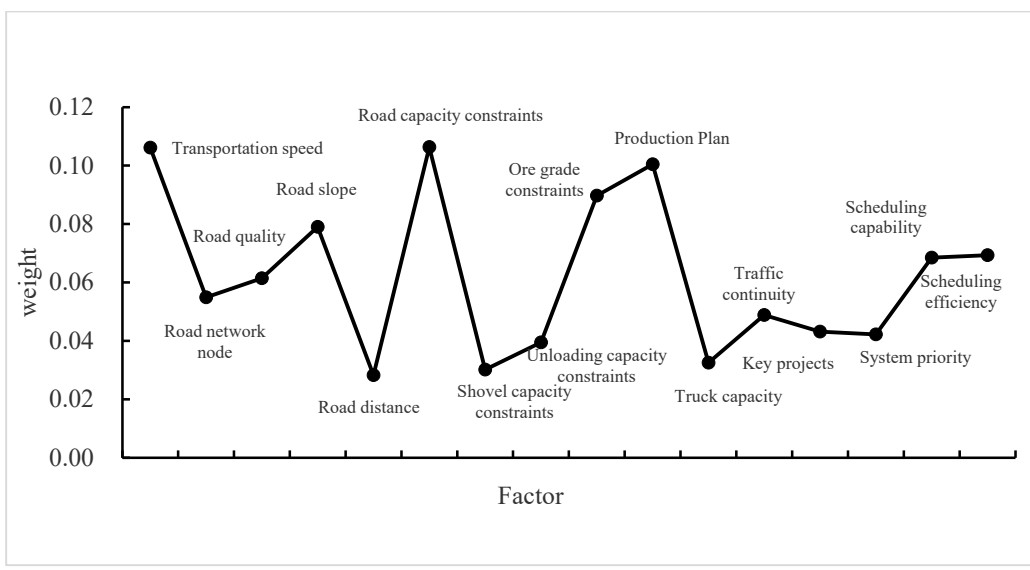

**Figure 1.** Relative importance of each factor.

### 4.3. Comprehensive Evaluation of Truck Dispatching System in Open-Pit Mine Based on Improved GRA-TOPSIS Method

The weighted normalization matrix $R$ was obtained using Equation (17), according to which the optimal set of gray correlation factors $U_j^*$ was selected on the basis of the improved GRA-TOPSIS: 0.106, 0.055, 0.061, 0.079, 0.028, 0.106, 0.030, 0.039, 0.090, 0.100, 0.033, 0.049, 0.043, 0.042, 0.068, and 0.069. Taking $U_j^*$ as the reference sequence, the pros and cons of each truck dispatching system were evaluated.

Using Equations (22)–(25), the Euclidean distances between the five truck dispatching systems ($E_1$, $E_2$, $E_3$, $E_4$, and $E_5$) and the positive and negative ideal solutions were calculated: $d_1^+ = 1.087$, $d_2^+ = 1.388$, $d_3^+ = 1.739$, $d_4^+ = 1.855$, and $d_5^+ = 1.151$; $d_1^- = 1.651$, $d_2^- = 1.004$, $d_3^- = 0.838$, $d_4^- = 0.494$, and $d_5^- = 1.449$. The relative closeness of the gray correlation was obtained using Equation (26): $G_1 = 0.6031$, $G_2 = 0.4198$, $G_3 = 0.3254$, $G_4 = 0.2103$, and $G_5 = 0.5573$. Accordingly, the five truck dispatching systems could be ranked from good to bad as $E_1 > E_5 > E_2 > E_3 > E_4$.

In order to facilitate the comparison and analysis of the results, the closeness degree based on the traditional TOPSIS method was calculated using Equations (14)–(20). According to the literature [20], the gray correlation degree of the truck dispatching system was calculated using the GRA method. Finally, the extreme values and coefficients of variation under each evaluation model were calculated, and the results are shown in Table 6. The comparison results are shown in Figure 2.

**Table 6.** Evaluation results of each model.

| Mine | GRA-TOPSIS | | GRA | | TOPSIS | |
|---|---|---|---|---|---|---|
| | Final Results | Sequence | Final Results | Sequence | Final Results | Sequence |
| $K_1$ | 0.6031 | 1 | 0.8408 | 1 | 0.6191 | 1 |
| $K_2$ | 0.4198 | 3 | 0.6927 | 3 | 0.5189 | 3 |
| $K_3$ | 0.3254 | 4 | 0.6168 | 4 | 0.3464 | 4 |
| $K_4$ | 0.2103 | 5 | 0.5511 | 5 | 0.3368 | 5 |
| $K_5$ | 0.5573 | 2 | 0.7875 | 2 | 0.6108 | 2 |
| Extreme value | 0.3927 | | 0.2898 | | 0.2821 | |
| Coefficient of variation | 0.1622 | | 0.1189 | | 0.1379 | |

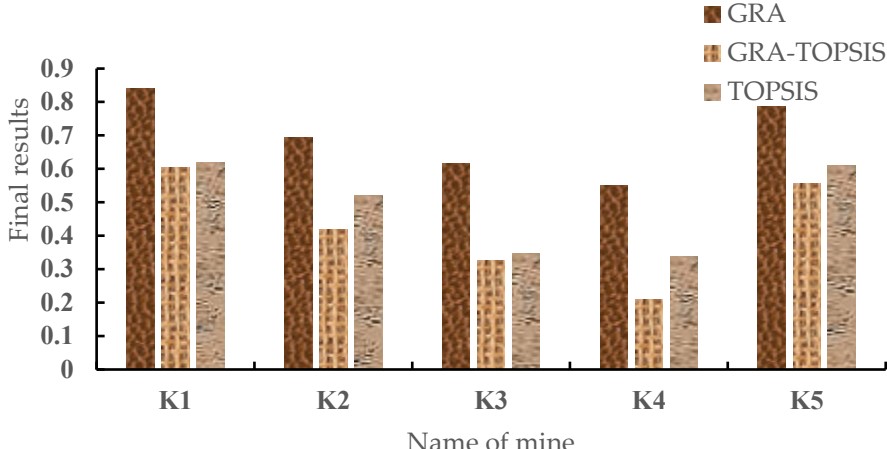

**Figure 2.** Comparison of evaluation model results.

According to Table 6, using the GRA-TOPSIS evaluation model, the truck dispatching systems of the five mines were ranked from good to bad as $E_1 > E_5 > E_2 > E_3 > E_4$. Using the GRA and TOPSIS evaluation models, the truck dispatching systems of the five mines were ranked in the same order. Thus, the calculation results of the three evaluation models were highly consistent, indicating that the established GRA-TOPSIS evaluation model had good adaptability in the evaluation of truck dispatching systems in open-pit mines. Using the GRA-TOPSIS, GRA, and TOPSIS methods, the extreme values were 0.3927, 0.2898, and 0.2821, respectively, while the coefficients of variation were 0.1622, 0.1189, and 0.1379, respectively. A larger extreme value and a larger coefficient of variation enable a greater dispersion degree of the comprehensive evaluation value of the truck dispatching system in the open-pit mine, improving the distinction between the pros and cons of each system. The extreme value and coefficient of variation based on the GRA-TOPSIS method were the largest, indicating its superior distribution of results and more obvious comprehensive difference between each system, enabling an intuitive analysis of the comprehensive application level of each system. When using the GRA or TOPSIS methods, the difference between the extreme values and the coefficients of variation was small, and the resolution level was not high. This can lead to ambiguous evaluation results and low accuracy. In summary, the comprehensive evaluation model of GRA-TOPSIS established in this paper has more prominent advantages and stronger adaptability in analyzing and processing the advantages and disadvantages of truck dispatching systems, reflecting this study's value.

According to Figure 2, the truck dispatching system of mine $K_1$ was the best, while that of $K_4$ was the worst. The closeness calculation result based on the GRA-TOPSIS method was located between the results of GRA and TOPSIS, indicating that the method combined the advantages of both models, taking into account the integrity and correlation of the evaluation. This verifies the rationality of the application of the GRA-TOPSIS comprehensive evaluation model.

### 4.4. Difference Analysis of Truck Dispatching System Based on Radar Chart

In order to more intuitively compare the advantages and disadvantages of the truck dispatching systems of the five open-pit mines as a function of different factors and aspects, the gray correlation degree of the five dispatching systems in terms of optimal route, traffic flow planning, and real-time dispatching was calculated. The results are shown in Table 7. According to the degree of closeness, the pros and cons of each subsystem were ranked and compared using the radar analysis method, as shown in Figure 3.

**Table 7.** Evaluation results of each subsystem.

| Name of Mine | $G_i$ (Total Closeness) | Sequence | Criterion Layer Closeness | | | | | |
|---|---|---|---|---|---|---|---|---|
| | | | Route Optimization | Sequence | Traffic Flow Planning | Sequence | Real-Time Dispatching | Sequence |
| $K_1$ | 0.6031 | 1 | 0.5589 | 1 | 0.6621 | 1 | 0.6380 | 1 |
| $K_2$ | 0.4198 | 3 | 0.5096 | 3 | 0.3332 | 3 | 0.5596 | 2 |
| $K_3$ | 0.3254 | 4 | 0.4441 | 5 | 0.2602 | 4 | 0.3872 | 4 |
| $K_4$ | 0.2103 | 5 | 0.4671 | 4 | 0.1862 | 5 | 0.2575 | 5 |
| $K_5$ | 0.5573 | 2 | 0.5197 | 2 | 0.6059 | 2 | 0.5378 | 3 |

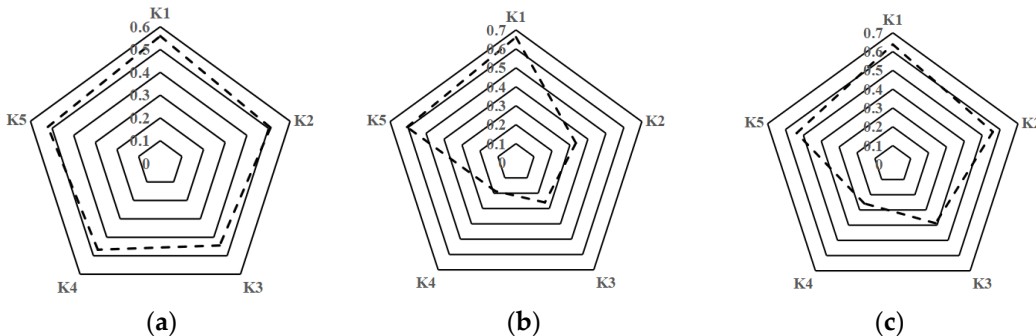

**Figure 3.** Radar diagram of each subsystem: (**a**) route optimization; (**b**) traffic flow planning; (**c**) real-time dispatching.

## 5. Conclusions

(1) In order to improve and balance the factor weight calculation, thus avoiding an invalid final evaluation result due to unreliable weight calculation, this paper adopted the G1 method, the improved CRITIC method, and game theory to calculate the subjective weights, objective weights, and final weights of the factors, resulting in a more reasonable weighting.

(2) Considering that the evaluation factors of the truck dispatching system in open-pit mines are not only incompatible, but also mainly qualitative, the TOPSIS method was introduced to evaluate the system. The traditional TOPSIS method was improved using the GRA method, and a comprehensive evaluation model was established. The truck dispatching systems of five open-pit mines were taken as the study objects. In addition, blind number theory was introduced to process the expert scoring results, thereby reducing the influence of subjective factors on their reliability. The results showed that the established GRA-TOPSIS model could improve the resolution level. The model realizes the exact sorting of the overall advantages and disadvantages of the existing truck dispatching system. The model can also study the advantages and disadvantages of each truck dispatching system subsystem, and analyze the differences between the systems according to the calculation results, providing a theoretical basis for the optimization and upgrading of the truck dispatching system. Satisfactory results were obtained, verifying its applicability and providing a scientific and effective method for system evaluation.

(3) In the evaluation of truck dispatching systems in open-pit mines, not only are there many influencing factors, but the relationship between these factors is also complex. Therefore, in follow-up research, it is necessary to further improve the factor system structure and optimize the factor classification standard to improve the applicability of the model. In addition, based on the comprehensive evaluation model of the TOPSIS method, the rationality of the factor quantification determines the reliability of the evaluation results. In this paper, blind number theory is used to process the factor data, and the calculation process is relatively complicated. Therefore, further

optimizing the factor data quantification method is also one of the key directions of future research.

**Author Contributions:** Conceptualization and investigation, X.K.; methodology, X.X.; software and writing—review and editing, Y.Z.; data curation, formal analysis, and validation, Q.K.; resources and supervision, Q.L. All authors have read and agreed to the published version of the manuscript.

**Funding:** This research was funded by the National Key Research and Development Plan, grant number 2019YFC0605305, Ministry of Science and Technology of the People's Republic of China.

**Institutional Review Board Statement:** Not applicable.

**Informed Consent Statement:** Not applicable.

**Data Availability Statement:** Not applicable.

**Conflicts of Interest:** The authors declare no conflict of interest.

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
