# Peer review of "Research on Comprehensive Evaluation Model of a Truck Dispatching System in Open-Pit Mine"

_sustainability, doi:10.3390/su14159062_

Round 1

Reviewer 1 Report

This paper presents a comprehensive evaluation factor model of a truck dispatching system in open-pit mine. Three dimensions are elaborated for building the model: optimal route, traffic flow planning and real-time dispatching. Game theory is also exploited for the model.

Terms such as “at present have to be avoided in the paper. The format of References has to be respected : for instance sometime as in line 70 or in line 73.

The introduction has to be improved by detailing the problem and the necessity of evaluation factor model. It is important to structure the introduction and make it more clear. After the context, a presentation of the problem, the exposition of how the problem is solved in the literature and how it will be in this paper, and finally the presentation of the section of the paper.

There is no literature review in the paper! It is possible if this is included in the introduction. It seems that it is not the case. The section 2 could become the third or the introduction could be more detailed for including the literature review.

In the section 3, the improved GRA-TOSIS method has been presented. But no detailed explanation before on the GRA method. A sub-section and a paragraph could be added for increasing the paper quality.

At the end of the section 4, may be a synthesis on the real interest of the paper proposition will also improve the quality of this paper. The objectives have to be clearly defined before (in the introduction) and this part will allow to show how these objectives have been attained.

This paper will be good after improvements.

Author Response

Point 1: Terms such as “at present” have to be avoided in the paper. The format of References has to be respected : for instance sometime as in line 70 or in line 73.

Response 1: “at present” has been deleted. The format of References has been changed.

Point 2: The introduction has to be improved by detailing the problem and the necessity of evaluation factor model. It is important to structure the introduction and make it more clear. After the context, a presentation of the problem, the exposition of how the problem is solved in the literature and how it will be in this paper, and finally the presentation of the section of the paper.

Response 2: This paper has expounded the necessity of establishing a evaluation factor model, such as "Through comparison, it was found that several studies have been conducted on the opti-mization of the truck dispatching system, while only few studies have been conducted on the comprehensive evaluation of the truck dispatching system. Therefore, a relatively ma-ture theoretical system, a perfect evaluation factor system, and an effective comprehensive evaluation model have not yet been formed, making it difficult to truly guide practice.” In view of the necessity of establishing the evaluation factor model, some contents have been added:

“The optimization of the truck dispatching system in open pit mines mainly focuses on 3 aspects: route optimization, traffic flow planning and real-time dispatching. Besides, the influencing factors involved in each optimization aspect are different. Therefore, on the basis of in-depth research on the theory and process of truck scheduling in open pit mines, this paper takes three aspects of optimal route, traffic flow planning and real-time dispatching into consideration. A comprehensive evaluation model for truck dispatching systems in open-pit mines.”

In view of the problems existing in the evaluation of the current truck dispatching system, this paper establishes an evaluation based on GRA-TOPSIS, introduces the game theory to calculate the index weight, and adopts the blind number theory to realize the quantitative processing of qualitative factors. The purpose of establishing the model has been added:

" Through this evaluation model, the existing truck dispatching system can be evaluated. Besides, the existing system can be optimized, upgraded and improved according to the evaluation results, which can further improve the production efficiency and management level of open-pit mining enterprises, and further control the operating costs of enterprises."

Point 3: There is no literature review in the paper! It is possible if this is included in the introduction. It seems that it is not the case. The section 2 could become the third or the introduction could be more detailed for including the literature review.

Response 3: References and citations in this article are marked.

In this paper, the establishment of the evaluation factor system is introduced in section 1, the theoretical basis of the established model is introduced in section 2, the established model is verified with an example in section 3. The structure of the article is reasonable and orderly.

Point 4: In the section 3, the improved GRA-TOSIS method has been presented. But no detailed explanation before on the GRA method. A sub-section and a paragraph could be added for increasing the paper quality.

Response 4: Because of the limited space of this paper, and GRA as a way to improve TOPSIS, the focus is on the calculation principle of the improved model. Therefore, only basic principles of the GRA mathematical model has been added in this paper:

" GRA is one of the core contents of grey system theory. It can describe and evaluate the state of the system over a period of time semi-qualitatively and semi-quantitatively by analyzing the known information, and then evaluate the overall level of the system. it is especially suitable for the evaluation environment with less data and fuzzy gray information.”

Point 5: At the end of the section 4, may be a synthesis on the real interest of the paper proposition will also improve the quality of this paper. The objectives have to be clearly defined before (in the introduction) and this part will allow to show how these objectives have been attained.

Response 5: A synthesis on the real interest of the paper proposition has been added:

" The model realizes the exact sorting of the overall advantages and disadvantages of the existing truck dispatching system. The model can also study the advantages and disad-vantages of each truck dispatching system subsystem, and analyze the differences be-tween the systems according to the calculation results, providing a theoretical basis for the optimization and upgrading of the truck dispatching system. and obtain sSatisfactory re-sults were obtained, verifying its applicability and providing a scientific and effective method for system evaluation."

Reviewer 2 Report

The authors propose an evaluation model based on GRA-TOPSIS was established in order to improve the comprehensive evaluation of the truck dispatching system. The paper is well structured and it is of interest for the Journal’s readership. Although the proposed approach is well designed, it is unclear how uncertainties are taken into consideration in evaluation. The results presented in the paper provide single values and not a range of them. No sensitivity analysis is conducted, and it should be performed.

Author Response

Point 1: The authors propose an evaluation model based on GRA-TOPSIS was established in order to improve the comprehensive evaluation of the truck dispatching system. The paper is well structured and it is of interest for the Journal’s readership. Although the proposed approach is well designed, it is unclear how uncertainties are taken into consideration in evaluation. The results presented in the paper provide single values and not a range of them. No sensitivity analysis is conducted, and it should be performed.

Response 1:  Aiming at the difficulty of quantifying qualitative indicators, this paper introduces the blind number theory. Considering the influence of expert subjectivity, this paper introduces the concept of "expert credibility". When assigning the factor, the expert gives the factor score interval, which can consider the dynamic variability of the truck dispatching system and the uncertainty of the expert in the evaluation process. Finally, the factor score is processed based on the blind number theory, not a single value.

Reviewer 3 Report

In this article, author shares a model on evaluation model of truck
dispatching system in open pit mine based on combination weighting method of the game theory. The topic is interesting. Some of the comments to improve the paper are as follows:

1. Use of the full term rather than acronyms or chemical formulas in the title and in the abstract.

2. The title is not clear, appealing, interesting and specific. I suggest to revise the paper title to make it more concise and suitable.

3. Re-edit Eqs. (18) by using MathType.

4. The benefits of the proposed method have been demonstrated clearly. What’s the limitation of the method? Are there other ways that the results can be further improved? One or two remarks should be given to discuss it in detail.

***

Author Response

Point 1: Use of the full term rather than acronyms or chemical formulas in the title and in the abstract.

Response 1: The full term has been added in the abstract.

Point 2: The title is not clear, appealing, interesting and specific. I suggest to revise the paper title to make it more concise and suitable.

Response 2: The title has been changed into “Research on Comprehensive Evaluation Model of a Truck Dispatching System in Open-Pit Mine”

Point 3: Re-edit Eqs. (18) by using MathType.

Response 3: All the Eqs have been checked and re-editted by using Math Type.

Point 4: The benefits of the proposed method have been demonstrated clearly. What’s the limitation of the method? Are there other ways that the results can be further improved? One or two remarks should be given to discuss it in detail.

Response 4: Some contents are added in this paper:

"In the evaluation of truck dispatching systems in open-pit mines, not only are there many influencing factors, but the relationship between these factors is also complex. Therefore, in follow-up research, it is necessary to further improve the factor system structure and optimize the factor classification standard to improve the applicability of the model. In addition, based on the comprehensive evaluation model of the TOPSIS method, the ra-tionality of the factor quantification determines the reliability of the evaluation results. In this paper, the blind number theory is used to process the factor data, and the calculation process is relatively complicated. Therefore, further optimizing the factor data quantification method is also one of the key directions of future research.”